# Focus of Attention Improves
# Information Transfer in Visual Features

**Matteo Tiezzi**[1], **Stefano Melacci**[1],[*] **Alessandro Betti**[1], **Marco Maggini**[1], **Marco Gori**[1,2]

[1]DIISM, University of Siena, Siena, Italy
[2]Maasai, Universitè Côte d'Azur, Nice, France
{mtiezzi,mela}@diism.unisi.it,alessandro.betti2@unisi.it,
{maggini,marco}@diism.unisi.it

## Abstract

Unsupervised learning from continuous visual streams is a challenging problem that cannot be naturally and efficiently managed in the classic batch-mode setting of computation. The information stream must be carefully processed accordingly to an appropriate spatio-temporal distribution of the visual data, while most approaches of learning commonly assume uniform probability density. In this paper we focus on unsupervised learning for transferring visual information in a truly online setting by using a computational model that is inspired to the principle of least action in physics. The maximization of the mutual information is carried out by a temporal process which yields online estimation of the entropy terms. The model, which is based on second-order differential equations, maximizes the information transfer from the input to a discrete space of symbols related to the visual features of the input, whose computation is supported by hidden neurons. In order to better structure the input probability distribution, we use a human-like focus of attention model that, coherently with the information maximization model, is also based on second-order differential equations. We provide experimental results to support the theory by showing that the spatio-temporal filtering induced by the focus of attention allows the system to globally transfer more information from the input stream over the focused areas and, in some contexts, over the whole frames with respect to the unfiltered case that yields uniform probability distributions.

## 1 Introduction

Nowadays the most popular benchmarks in the machine learning community are composed of batches of data that are commonly processed in an offline manner using stochastic updates of the model parameters, periodically shuffling the available samples [23, 17, 8]. A smaller effort has been devoted by the research community to the direction of focusing on a single, potentially life-long video, in which the model continuously processes a stream of frames, that is a very natural setting resembling the flow of information that hits the eyes of each human [5]. An important feature of the human visual system that is frequently neglected in several algorithms is the attention mechanism that drives the gaze over different spatial regions of the input stimulus. As a matter of fact, it is implicitly assumed that all the pixels equally contribute to the learning process, assuming a uniform probability distribution of their coordinates over the retina. In the last few years, a lot of importance has been devoted to attention in neural models, for example in learning to play games [36], in learning task-specific attention [21], or in mixing bottom-up and top-down attention [32]. A different research direction, closer to Neuroscience, is the one that specifically studies saliency in the context of the human visual attention systems [6], where dynamic models of visual attention have been recently proposed, able to predict in an online manner the trajectory of the attention [33, 34].

---

[*]Corresponding author.

In this paper, we cast the problem of processing a visual stream in a truly online setting, motivated by recent studies that connected learning over time and classical mechanics [5, 3, 4]. The framework proposed in [5] naturally deals with learning problems in which time plays a crucial role, and it is well-suited to learn from streams of visual data in a principled way. The temporal trajectories of the variables of the learning problem are modeled by the so called 4th order Cognitive Action Laws (CALs) that come from stationarity conditions of a functional, as it happens for generalized coordinates in classical mechanics. We intersect these ideas with the recent human-like attention model of [34], that has shown state-of-the art results in focus estimation. Motion and visual features are treated as a mass distribution in the gravitational field that determines the trajectory of the focus of attention. The focus of attention implements a filtering procedure on the input video, allowing the system to deal only with those areas that would attract the human attention. We propose a 2nd order model that, under some mild conditions, leads to a simplified and more manageable instance of the CALs, yielding ODEs of same order of the ones that drive the attention.

With the goal of studying the impact of the focus of attention dynamics in videos, we consider the problem of transferring information from the input visual stream to the output space of a neural architecture that performs pixel-wise predictions [3, 4]. This problem consists in maximizing the Mutual Information (MI) index [5]. One of the key issues with MI maximization over time, especially when focusing the attention on a few pixels, is the fact that stochastic updates of the model parameters do not keep track of the entropy of the output space due to the data processed so far, leading to poorly informed updates. We investigate the case in which the global changes in the entropy of the output space are approximated by introducing a specific constraint or a moving average. It turns out that, when learning over the focus trajectory, the MI index grows more significantly over the focused areas with respect to the unfiltered case, and, in some configurations, it is also larger than considering other distributions of the pixel coordinates. This suggests that filtering the information by a bottom-up attention model helps the system in transferring information from the whole stream.

The topic of MI maximization has recently attracted the attention of several researches [2, 13, 29, 22, 31]. Most of the recent works are about customized MI-based criteria to learn representations for downstream tasks, that is not the case of this paper. Moreover, [13, 29] are based on surrogate functions that loosely approximate [31] the continuous MI formulation, while here we directly consider the discrete MI index, that, for instance, has been previously used as criterion to relate different views of the input data [14] or in clustering [20]. The information transferred by multi-layer networks is discussed in the context of the popular information bottleneck principle by Naftali Tishby and other authors as a mean to study deep network internal dynamics [30, 27, 24]. Several of the topics that are covered by this work intersect the research activity of Karl Friston and of his co-authors, see, e.g., [9, 10, 25, 15]. This work is specifically focused on empirically evaluating the effects of a focus of attention mechanism in deep networks that are trained on video streams in an unsupervised manner.

In summary, the contributions of this paper are: (1) we study human-like attention mechanisms in conjunction with learning in video data, (2) considering a new 2nd order differential model and (3) evaluating the impact of different criteria to approximate the entropy estimate over the whole stream. This paper is organized as follows. Section 2 describes the learning framework, 2nd order models, and the problem of MI maximization. Section 3 is about injecting the focus of attention dynamics, while experiments are reported in Section 4. Section 5 concludes the paper with ideas for future work.

## 2   Learning over Time

We consider the problem of processing a stream of data over time and, in particular, a stream of video frames $t \mapsto u(t)$ from a target source, being $u(t)$ the frame at time $t$ in the time horizon $[0, T]$. The stream is processed by a neural network whose weights and biases at time $t$ are represented by the generic vector variable $w(t)$, while $\dot{w}(t), \ddot{w}(t)$ are respectively its first and second derivatives. Our work is rooted in the ideas presented in [5, 3, 4], where learning is described in analogy with classical mechanics, as a variational problem whose objective is to find a stationary point of the following functional $\Gamma^{(\xi)}$ of the maps $t \mapsto w(t) \in \mathbb{R}^n$,

$$\Gamma^{(\xi)}(w) := \int_0^T L(t, w(t), \dot{w}(t), \ddot{w}(t)) \, dt = \int_0^T h(t) \big( K(\dot{w}(t), \ddot{w}(t)) - \xi V(w(t), u(t)) \big) \, dt. \quad (1)$$

The Lagrangian $L$ is composed of a kinetic energy $K$ and a potential energy $V$, while $h(t)$, when appropriately chosen, is responsible of introducing energy dissipation. The term $\xi \in \{-1, 1\}$ is selected in function of the way $K$ is implemented (see [3] for details[2]). In particular, in [3, 4, 5] we have $\xi = -1$, $h(t) = e^{\theta t}$, $V$ is composed of the loss function $U$ of the considered problem and a quadratic regularizer on $w(t)$, and $K$ includes the squared norm of the derivatives plus their dot product, leading to

$$\Gamma^{(-1)}(w) \equiv \Gamma(w) = \int_0^T e^{\theta t} \left( \frac{\alpha}{2} |\ddot{w}(t)|^2 + \frac{\beta}{2} |\dot{w}(t)|^2 + \gamma \dot{w}(t) \ddot{w}(t) + \frac{k}{2} |w(t)|^2 + U(w(t), u(t)) \right) dt, \tag{2}$$

where $\theta \in \mathbb{R}$ and $\alpha$, $\beta$, $\gamma$, $k$ are custom positive scalars, $|\cdot|$ is the Euclidean norm in $\mathbb{R}^n$ and $\cdot$ is the standard scalar product in $\mathbb{R}^n$, being $n$ the size of $w(t)$.

The Euler-Lagrange (EL) equations of Eq. (2) yield the Cognitive Action Laws (CALs), 4th order differential equations that, when integrated, allows $w$ to be updated over time. In particular, they are[3]

$$\alpha w^{(4)} + 2\theta\alpha w^{(3)} + (\theta^2\alpha + \theta\gamma - \beta)\ddot{w} + (\theta^2\gamma - \theta\beta)\dot{w} + kw + \nabla U(w, u) = 0, \tag{3}$$

being $w^{(4)}$ and $w^{(3)}$ the fourth and third derivatives of $w$, respectively, and $\nabla U$ is the gradient of $U$ with respect to its first argument. Cauchy's initial conditions can be provided on $w$ and $\dot{w}$, while stationarity conditions of $\Gamma$ prescribe that Eq. (3) must be paired with boundary conditions on the right border ($t = T$). Thus, in order to solve the problem of determining $w(t)$ in a causal way (i.e. in such a way that the solution $w$ at time $t$ does not depend on values in $(t, T]$), the fulfilment of the boundary conditions in $t = T$ is approximated in [3] by introducing a mechanism that sets ("resets") to zero all the derivatives up to $w^{(3)}$, whenever their norms become too large. See [3] for more details on CALs.

## 2.1 Second-Order Laws

Despite their robust principled formulation, the main drawbacks of the 4th order CALs is the difficulty in tuning the parameters that weigh the contribute of the derivatives, and the computational/memory burden due to the integration of a 4th order ODE. Moreover, the theoretical guarantees on the stability of Eq. (3) are experimentally shown to not be necessarily needed, mostly due to the aforementioned derivative reset procedure [3]. For these reasons, in this paper we will use the CAL theory in a particular *causal* regime of the parameters for which two important simplifications are attained. First, the dynamics of the weights are described by a 2nd order ODE (instead of Eq. (3)). Second, we get direct causality without the need of any reset mechanisms.

The limiting procedure that leads to the 2nd order laws is based on a conjecture by De Giorgi [1] which has been subsequently proved and studied in [28, 26, 18]. In detail, we consider a reparametrization in terms of $\varepsilon > 0$ of the $\Gamma$ functional, where $\theta \to -1/\varepsilon$, $\alpha \to \varepsilon^2\alpha$, $\beta \to \varepsilon\beta$. This allows us to rewrite Eq. (2) in line with De Giorgi's functional,

$$\Gamma_\varepsilon(w) := \int_0^T e^{-t/\varepsilon} \left( \frac{\alpha\varepsilon^2}{2} |\ddot{w}(t)|^2 + \frac{\beta\varepsilon}{2} |\dot{w}(t)|^2 + \frac{k}{2} |w(t)|^2 + U(w(t), u(t)) \right) dt, \tag{4}$$

where we also chose, for simplicity, $\gamma = 0$. Letting $\varepsilon \to 0$, the minima of the functional $\Gamma_\varepsilon$ with fixed initial conditions on $w$ and $\dot{w}$ converges to the solution of a Cauchy problem based on a 2nd order differential equation, thus gaining full causality, i.e., $\varepsilon$ measures the "degree of causality" of the solution. Notice that the factor $e^{-t/\varepsilon}$ in Eq. (4) becomes peaked on $t = 0$ as $\varepsilon \to 0$, and the minimization procedure of $\Gamma_\varepsilon$ will be mainly concerned in the minimization of the loss calculated at $t = 0^+$. At a first glance, this might seem counter-intuitive. However, it becomes a useful feature when considered in conjunction with the properties of the input signal $u(t)$. Let us indicate with $\tau > 0$ the temporal scale of $u(t)$, that is a small time span under which the variations of $u(t)$ are semantically negligible. The whole temporal interval $[0, T]$ can be partitioned into $\lceil T/\tau \rceil$ disjoint intervals $[0, \tau), [\tau, 2\tau), \ldots [(\lceil T/\tau \rceil - 1)\tau, T)$, in each of which the aforementioned picky behaviour is not critical due to the temporal scale of $u(t)$. The minimization of Eq. (4) can be iteratively defined by minimizing $\Gamma_\varepsilon$ in each interval, where the conditions on the left boundary are given by the solution

of the minimization in the previous interval. When $\varepsilon \ll \tau$, the minimization problem can be well interpreted in terms of the value of $U(\cdot, u(\kappa\tau))$, for $\kappa = 0, \ldots, \lceil T/\tau \rceil - 1$.

To introduce the EL equations of the newly introduced problem, for simplicity, we will describe the limiting procedure in the interval $[0, T]$, that applies to each of the $\lceil T/\tau \rceil$ previously described intervals. The EL equations for the minimizer of $\Gamma_\varepsilon$ with initial conditions $w(0) = w^0$ and $\dot{w}(0) = w^1$ are

$$\begin{cases} \varepsilon^2 \alpha w^{(4)}(t) - 2\varepsilon\alpha w^{(3)}(t) + (\alpha\epsilon^2 - \varepsilon\beta)\ddot{w}(t) + \beta\dot{w}(t) + kw(t) + \nabla U(w(t), u(t)) = 0; \\ w(0) = w^0, \quad \alpha\dot{w}(0) = \alpha w^1, \quad \alpha\ddot{w}(T) = 0, \quad \alpha\varepsilon w^{(3)}(T) = \beta\dot{w}(T), \end{cases} \tag{5}$$

and the following theorem holds:

**Theorem 1.** *The solution of the problem* (5) *converges (weakly in* $H^1((0,T), \mathbb{R}^n)$ *) to the solution of*

$$\begin{cases} \alpha\ddot{w}(t) + \beta\dot{w}(t) + kw(t) + \nabla U(w(t), t) = 0; \\ w(0) = w^0, \qquad \dot{w}(0) = w^1. \end{cases} \tag{6}$$

Equations (6) are 2nd order CALs. They are simpler than the 4th order CALs of Eq. (3), even if they maintain their principled nature. See the supplementary material for formal proofs and further details.

## 2.2 Mutual Information in Video Streams

We consider the problem of transferring information from an input visual stream to the output space of a multi-layer convolutional network with $\ell$ layers, that processes each frame and yields $m$ *pixel-wise* predictions. This corresponds to the maximization of the Mutual Information (MI) from the pixels of the input frames to the $m$-dimensional output space yielded by the $m$ units of the last layer, being $m$ the size of the filter bank in layer $\ell$. Hyperbolic tangent is used as activation function in each layer $j < \ell$, while the last layer is equipped with a softmax activation, generating $m$ probabilities $p(w, x, u) = (p_1(w, x, u), \ldots, p_m(w, x, u))$, being $x$ a pair of pixel coordinates and $u$ the processed frame. This problem is studied in [5] and related papers [4, 3], where single-layer models (or stacks of sequentially trained single-layer models) are considered, while, in this paper, we exploit a deep network trained end-to-end. Previous approaches based on kernel machines can be found in [12, 11].

In order to define the MI index, we consider a generic, time independent weight configuration $\omega \in \mathbb{R}^n$. We introduce the average output activation on the video portion between time instants $t_1$ and $t_2$,

$$P(\omega, t_1, t_2) \equiv \int_{t_1}^{t_2} \overline{P}(\omega, t)dt := \int_{t_1}^{t_2} \int_R p(\omega, x, u(t))\mu(x, t)dxdt, \tag{7}$$

where $\mu(x, t)$ is a *spatio-temporal* density and $R$ is the set of points that constitute the retina. The MI index over the video portion $[t_1, t_2]$, is defined as

$$\begin{aligned} I(X, Y; \omega; t_1, t_2) &= -H(Y|X; \omega; t_1, t_2) + H(Y; \omega; t_1, t_2) \\ &= -\sum_{j=1}^m \int_{t_1}^{t_2}\!\!\int_X p_j(\omega, x, u(t)) \log p_j(\omega, x, u(t)) \, \mu(x, t)dxdt + \sum_{j=1}^m P_j(\omega, t_1, t_2) \log P_j(\omega, t_1, t_2) \end{aligned} \tag{8}$$

where $H$ is the entropy function, and $X$ and $Y$ are random variables ($Y$ is discrete) associated with the input[4] and output space, respectively[5]. When no further information is available, $\mu$ is commonly assumed to be uniform in time and space and it is normalized such that $\int_{t_1}^{t_2} \int_R \mu(x, t)dxdt = 1$.

Performing maximum-MI-based online learning of $w$ using the CALs in the time horizon $[t_1 = 0, t_2 = T]$ is not straightforward. Once we restore the dependency of $w$ on time, by inserting $w(t)$ in place of $\omega$, we cannot simply plug (minus) the MI index as a potential loss $U$ in the Lagrangian due to the lack of temporal locality. As a matter of fact, in order to implement online learning dynamics, $U$ must be temporally local, i.e., it should depend on $w$ and $u$ at time $t$ only. For this reason, the authors of [5] compute the MI index at time $t$, and not in an interval; the approximation of the MI in $[0, T]$ is yielded by the outer integration in the functional of Eq. (4) (or, equivalently, in the one of

Eq. (2)). A drawback of this formulation is that, due to this temporal assumption, it could lead to a loose approximation of the original term $H(Y; \cdot; \cdot, \cdot)$ of Eq. (8), for which the inner integration on time (Eq. (7)) is lost, and replaced by the outer integration of the functional. In order to better cope with the optimization dynamics, the two entropy terms are commonly weighted by positive scalars $\lambda_c$, $\lambda_e$. In addition to the plain-vanilla case we just described (referred to as PLA), we explore two other alternative criteria to mitigate the impact of time locality, that we will evaluate in Section 4. The first one (VAR) consists in introducing an additional auxiliary variable $s(t)$, that is used to replace $P$ of Eq. (7), while its variation, $\dot{s}(t)$, is constrained to be almost equivalent to $\overline{P}(w(t), t)$. The Lagrangian is augmented with $\lambda_s |\dot{s}(t) - \overline{P}(w(t), t)|^2$, a soft-constraint that enforces $s(t)$ to approximate the case in which the probability estimate is not limited to the current frame ($\lambda_s > 0$).[6] This idea is presented in [5] but not followed-up in any experimentation. As a second criterion (AVG), we propose to replace $P$ with the outcome $\nu$ of an averaging operation that keeps track of the past activation of the output units, i.e., $\nu(t) = \zeta_s \nu(t') + (1 - \zeta_s) \overline{P}(w(t), t)$, for two consecutive time instants $t > t'$.

## 3 Focus of Attention

The way video data is commonly processed by machines usually lacks a key property of the human visual perception, that is the capability of exploiting eye movements to perform shifts in selective visual attention. High visual acuity is restricted to a small area in the center of the retina (fovea), and the purpose of the Focus Of Attention (FOA) is to selectively orient the gaze toward relevant areas with high information, filtering out irrelevant information from cluttered visual scenes [19, 16, 34].

In the context of Section 2.2, we consider a visual stream and a neural architecture with $m$ output dimensions (per pixel), and we aim at developing the network weights $w$ such that the MI index is maximized as strongly as possible with respect to the model capacity. Of course, restricting the attention to a subset of the spatio-temporal coordinates of the video, due to a FOA mechanism, seems to inherently carry less information than when considering the whole video. However, in the latter case, the processed data will be characterized by a larger variability, mixing up noisy/background information with what could be more useful to understand the video. Such mixture of data could be harder to disentangle by a learning model than well-selected information coming from a human-like FOA trajectory, leading to a worse MI estimate. Curiously, the learning process restricted to the FOA trajectory could end-up in facilitating the development of the weights, so that the MI computed on the whole frame area could be larger than when learning without restrictions. Following the notation of Eq. (8), the MI maximization, for each $t$, is based on the spatial distribution $\mu(x, t)$. Such distribution models the relevance of each coordinate $x$ when learning from frame $u(t)$. In [3, 4], $\mu(x, t)$ is assumed to be uniform over the frame area, while in [5] it is also described the idea of considering $\mu$ ($f$ in [5]) as the most natural candidate for implementing a FOA-based mechanism. Let us assume that $a(t)$ are the spatial coordinates of the FOA at time $t$, then we define

$$\mu(x, t) := g(x - a(t)), \tag{9}$$

being $g$ a function that is peaked on $a(t)$. Following this parametrization of $\mu$, we borrow a state-of-the art model for scanpath $a(t)$ prediction defined in [34], that shares a physics-inspired formulation as CALs.[7] Such FOA model has been proven to be strongly human-like in free-viewing conditions [35]. It is based on the intuition that the attention emerges as a gravitational process, in which both low-level (gradient, contours, motion) or high-level features (objects, context) may act as gravitational masses. In particular, given the gravitational field $E(t, a(t))$, the law that drives the attention is

$$\ddot{a}(t) + \rho \dot{a}(t) - E(t, a(t)) = 0, \tag{10}$$

that is indeed another 2nd order model as the one we proposed in Section 2.1 (see [34] for more details). The dissipation is controlled by $\rho > 0$, and the importance of each mass can also be tuned. Interestingly, Eq. (10) describes the dynamics of the FOA, and it is not based on pre-computed or given saliency maps. In this paper, following [34], we consider two basic (low-level) perceptive features as masses, the spatial gradient of the brightness and the strength of the motion field. The

trajectories simulated by the model show the same patterns of movement characteristic of human eyes: *fixations*, when the gaze remains still in a location of interest; *saccades*, rapid movement to reallocate attention on a new target; *smooth pursuit*, slow movements performed in the presence of a visual feedback with the purpose of tracking a stimulus.

Different choices on $g$ are possible. In Section 4 we will consider the extreme case in which $g(x - a(t))$ is a Dirac delta on the coordinates $a(t)$ (we will refer to it as FOA), so that $\mu(x, t)$ is essentially a mono-dimensional signal. A less extreme setting is the one in which $g$ is a squared window centered in $a(t)$ that covers a small fraction of the frame (FOAW), while the most-relaxed setting is when $g$ is simply uniform on the whole frame (UNI), i.e., $a(t)$ is not used.

## 4   Experimental Results

We evaluated the amount of information transferred from different video streams with 2nd order laws of Section 2.1, using multiple instances of the deep convolutional network described in Section 2.2. A PyTorch-based implementation can be downloaded as supplementary material.

**Models.** Architectures are referred to as S (Small), D (Deeper), DL (Deeper and with a Larger number of neurons), and they are based on $5 \times 5$ filters (except for the last layer – $7 \times 7$ filters), $\ell = 3$ (S) or $\ell = 7$ (D, DL) layers, and either $m = 10$ (S, D) or $m = 32$ (DL) filters in layer $\ell$. Networks S and D are composed of 20 filters in each hidden layer, while DL has 32 filters in each hidden layer. Following Section 3, we compared 3 potential terms based on 3 different input probability densities $\mu(x, t)$, named UNI, FOA, FOAW (uniform, foa-restricted, and foa-window-restricted, respectively – window edge is $15\%$ of the min frame dimension). For each of them, we tested the 3 criteria of Section 2.2 to extend the temporal locality, PLA, VAR, AVG (fully local, variable-based, average). We integrated the differential equations using the Euler method.

**Setting & Data.** We considered three visual streams with $105k$ frames each. The first $100k$ frames are the ones on which learning is performed, integrating the CALs. Then, the developed weights $w(T)$ are used to measure the MI index over the following $5k$ frames, directly applying the MI formulation of Eq. (8), i.e., $I(X, Y; w(T); 100000, 105000)$, that is what we report in the results of this section. For all the models, independently on the probability density used in their potentials, we measured the MI index using $\mu(x, t)$ in the UNI, FOA, FOAW cases. This means that, for example,

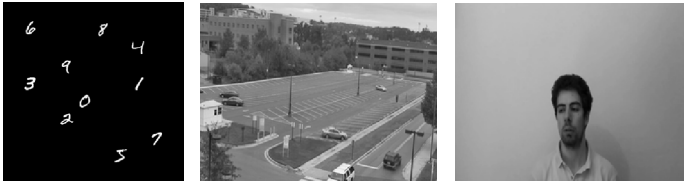

Figure 1: Sample frames taken from the SPARSEMNIST, CARPARK, CALL streams, left-to-right.

a model trained following the FOA trajectory is then evaluated in the 5k test frames either considering the whole frame area, the FOA trajectory, or the window-based FOA trajectory. The three streams (Fig. 1), have different properties. The first one, SPARSEMNIST, is composed of a static frame ($280 \times 280$) in which 10 digits from the MNIST data are sparsely located over a dark background. The second video, CARPARK, is taken from a fixed camera monitoring a car parking area in front of a building. The last video, CALL, is a recording taken from a webcam during a video call. Videos are repeated until the target number of frames is reached. The last two videos are processed at $240 \times 180$ pixels per frame, grayscale, $\approx 30$ frames per second. We selected videos that naturally represent repetitive contents, so that they can be repeated without introducing evident scene changes. Contents and events are heterogeneous among the videos.

**Parameters.** The FOA trajectory was generated by weighing the two gravitational masses 0.1 (frame details) and 1.0 (motion), respectively, and adjusting $\rho \in [0.1, 0.5]$ in order to adapt it to the each video. We analyze the behaviour of the FOA trajectories in Fig. 2. After a first experimentation in which we qualitatively observed the behaviour of the 2nd order laws, we set $\alpha = 0.01$, $\beta = 0.1$, $k = 10^{-8}$. For each model we considered multiple weighing schemes of the parameters $\lambda_c \in \{10, 100, 200, 1000\}$, $\lambda_e \in \{20, 200, 400, 2000, 4000\}$, $\lambda_s \in \{10, 100, 1000\}$,

Table 1: Main result. Mutual Information (MI) index in three video streams, considering three neural architectures (S, D, DL). Each column (starting from the third one) is about the results of network trained using an input probability densities taken from {UNI, FOA, FOAW}, and tested measuring the MI index in all the three density cases (labeled in column "Test").

| Stream | Test | S UNI | FOA | FOAW | D UNI | FOA | FOAW | DL UNI | FOA | FOAW |
|---|---|---|---|---|---|---|---|---|---|---|
| SparseMNIST | UNI | 0.017 | **0.112** | 0.078 | 0.004 | **0.144** | 0.020 | 0.012 | **0.132** | 0.026 |
| | FOA | 0.239 | **0.486** | 0.391 | 0.103 | **0.431** | 0.229 | 0.146 | **0.350** | 0.194 |
| | FOAW | 0.154 | **0.209** | 0.197 | 0.144 | **0.255** | 0.157 | 0.117 | **0.215** | 0.131 |
| Carpark | UNI | **0.776** | 0.601 | 0.695 | 0.653 | 0.556 | **0.745** | 0.445 | 0.292 | **0.496** |
| | FOA | **0.742** | 0.675 | 0.694 | 0.678 | 0.639 | **0.768** | 0.477 | 0.315 | **0.529** |
| | FOAW | **0.719** | 0.629 | 0.671 | 0.653 | 0.601 | **0.721** | 0.501 | 0.357 | **0.532** |
| Call | UNI | **0.329** | 0.314 | 0.315 | 0.339 | **0.556** | 0.350 | 0.208 | **0.304** | 0.218 |
| | FOA | **0.405** | **0.405** | 0.371 | 0.430 | **0.582** | 0.492 | 0.246 | **0.365** | 0.270 |
| | FOAW | **0.429** | 0.420 | 0.413 | 0.442 | **0.566** | 0.457 | 0.304 | **0.374** | 0.310 |

$\zeta_s \in \{0.01, 0.05, 0.07\}$, selecting the ones that returned the largest MI during the learning stage. As a general rule of thumb, using a lower value of the conditional entropy weighing term $\lambda_c$ w.r.t. the entropy weight $\lambda_e$, helps the model to exploit all the available output symbols. The network weights $w(0)$ were randomly initialized, enforcing the same initialization to all the compared model.

**Main result.** Our main results are highlighted in Tab. 1. Each column, starting from the third one, is about a model, defined by the pair (*architecture, density used in the training potential*). For each model, the MI index is reported when measured using different spatio-temporal densities (they are labeled in column "Test"). We used the temporal locality criterion that led to the best results. Overall, the models trained on FOA-based densities (columns FOA, FOAW) usually perform better than the ones that were exposed to a uniform $\mu(x, t)$ over the frame area (columns UNI). This is particularly noticeable in the SPARSEMNIST and CALL streams, characterized by a still and not-much-detailed background and few regions of interest, i.e. the digits or the moving speaker, respectively. The filtering approach induced by the attention in the training stage highly improves the information transfer over most of the considered test measurements, with just a few exceptions. These considerations holds at a lesser degree also in the CARPARK stream, in which frames are more detailed. The focus is attracted by a busy road or by people parking their cars. However, also the immediate surroundings of those regions contain much information, so that training with FOAW density achieves the best results in architectures D and DL, while the more extreme FOA approach do not compete with models trained considering the whole frame (UNI). In both the CARPARK and CALL streams, the S architecture does not benefit from learning over the attention trajectory. We

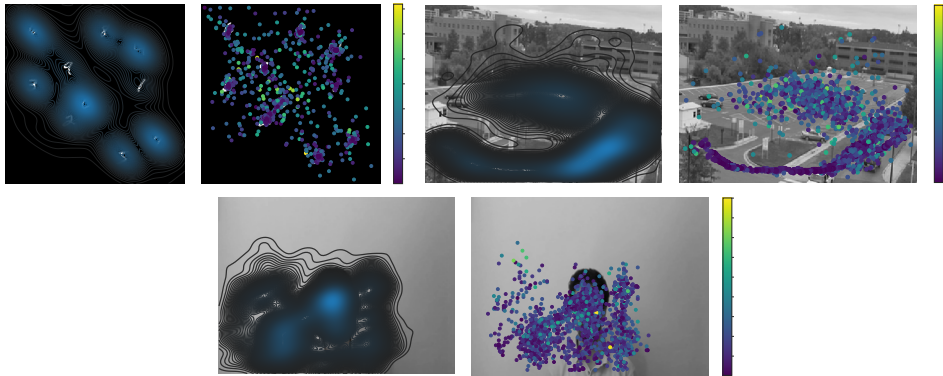

Figure 2: For each stream, we show (*left*) the areas mostly covered by FOA (blue: largest attention), and (*right*) the scatterplots of the fixation points, with hue denoting the magnitude of the FOA velocity (blue: slower; yellow: faster). Low-speed movements happen on the most informative areas (e.g., digits, busy roads, human presence/movement, respectively).

motivate this result by considering that S is a shallower model, that inherently learns lower level features that the other ones. These features are more common to different frame location, making the impact of attention less evident. In the case of SPARSEMNIST, the dark-uniform background dominates the frame, and learning over $a(t)$ induces a largest information transfer also in network S.

**Temporal locality.** In order to evaluate the impact of the temporal locality criteria (PLA, AVG, VAR), we restrict our analysis to models trained with a FOA-restricted probability density. In this case, we describe each model by the pair (*architecture*, *temporal locality criterion*), and we report results in Tab. 2. In general, the moving average criterion (AVG) achieves the best performances in all settings, with some exceptions. The CARPARK stream has temporal dynamics that are pretty repetitive and periodic (e.g., cars crossing the same crossroad etc.). Hence, the addition of a criterion to better keep track of the temporal information turns out to be less necessary. We notice higher value of MI index in the fully temporally local case (PLA) in architecture DL. This may be due to the fact that DL has a larger number parameters and units than the other nets, and it has intrinsically more capacity to memorize the temporal information. The MI index is lower that the one of the other architectures due to the largest size of the output space.

Table 2: Temporal locality. Mutual Information (MI) index in three video streams, considering three neural architectures (S, D, DL). Each column (starting from the third one) is about the results of network trained using the FOA trajectory with a temporal locality criterion taken from {PLA, AVG, VAR}, and tested measuring the MI index in all the three density cases (labeled in column "Test").

| | | S | | | D | | | DL | | |
|---|---|---|---|---|---|---|---|---|---|---|
| Stream | Test | PLA | AVG | VAR | PLA | AVG | VAR | PLA | AVG | VAR |
| SparseMNIST | UNI | 0.071 | **0.112** | 0.102 | 0.006 | 0.054 | **0.144** | 0.028 | **0.132** | 0.080 |
| | FOA | 0.425 | **0.486** | 0.298 | 0.149 | 0.321 | **0.431** | 0.119 | **0.350** | 0.184 |
| | FOAW | 0.183 | **0.209** | 0.208 | 0.146 | 0.184 | **0.255** | 0.127 | **0.215** | 0.176 |
| Carpark | UNI | **0.601** | 0.486 | 0.371 | 0.422 | **0.556** | 0.315 | **0.292** | 0.289 | 0.204 |
| | FOA | **0.675** | 0.521 | 0.401 | 0.458 | **0.639** | 0.326 | **0.315** | 0.307 | 0.209 |
| | FOAW | **0.629** | 0.548 | 0.447 | 0.489 | **0.601** | 0.389 | **0.357** | 0.357 | 0.277 |
| Call | UNI | 0.289 | **0.314** | 0.267 | 0.259 | **0.556** | 0.369 | **0.304** | 0.189 | 0.200 |
| | FOA | 0.326 | **0.405** | 0.265 | 0.328 | **0.582** | 0.459 | **0.365** | 0.214 | 0.260 |
| | FOAW | 0.383 | **0.420** | 0.373 | 0.368 | **0.566** | 0.443 | **0.374** | 0.274 | 0.275 |

**Random scanpaths.** We are left with the open question on whether the largest information transfer we experienced is due to the state-of-the art attention model we used or it is only due the reduction of the size of the input data. We compared models trained on the FOA trajectories used so far with the same networks trained randomly sampling $a(t)$ from a uniform distribution over the retina. The results of Fig. 3 show that the human-like trajectory estimated by the selected attention model has a clear positive impact in the information transfer. Interestingly, in the CARPARK case we sometimes observe that fixations which explore random coordinates highly foster information transfer. This confirms our previous statements regarding the large amount of information in whole the frame area.

**Learning dynamics.** We investigate the behaviour of the models during the training stage, in the case of architecture D and a single training/test probability density, FOA. The plots of Fig. 4, for each value $t$ of the $x$-axis, shows the MI index computed in the interval $[0, t]$ along the FOA trajectory, for different temporal criteria (PLA, AVG, VAR). The variable-based (VAR) model tends to quickly find a stationary condition of the estimated MI index value. Both PLA and AVG incur in an initial stage with evident fluctuations before becoming more stable, usually in larger values than VAR. The models have to deal with pretty varied conditions at the first stages of learning, which is limited to a single location in each frame. As long as time passes and a largest portion of stream is processed, fluctuations are mitigated reaching more stable configurations.

## 5 Conclusions and Future Work

In this work we delved into a novel approach to Mutual Information (MI) maximization rising from the conjunction of online entropy estimation mechanisms and human-like focus of attention. We introduced a 2nd order differential model, providing insightful experimental results to support the intuition that using the focus of attention to drive the learning dynamics fosters an increment of the

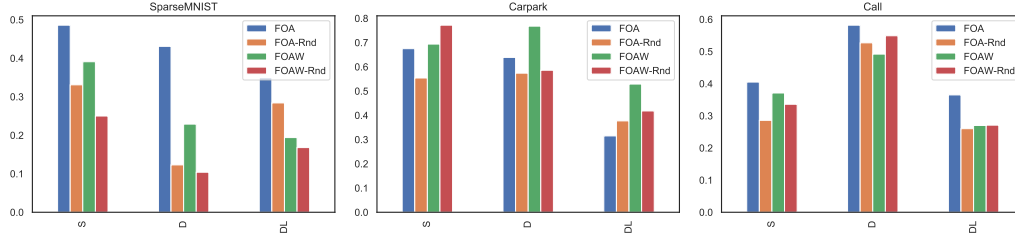

Figure 3: Comparison between models trained on a regular trajectory of the attention and on a random trajectory (suffix -RND), for architectures S, D, DL. Each bar is about a different training probability density, and the height of the bar is the test MI index along the regular FOA trajectory.

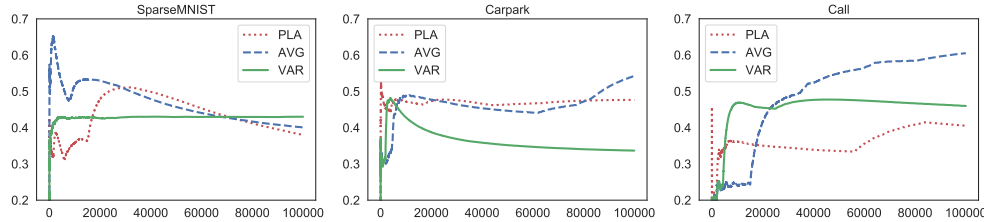

Figure 4: Learning dynamics (model D-FOA, different temporal criteria). The MI index is shown at different time instants. The index at time $t$ is evaluated along the FOA trajectory in the interval $[0, t]$.

globally transferred information from the input stream. Future work will be devoted to enforcing coherence over the predictions performed on the focus trajectory to develop high-level representations.

## Broader Impact

Our work is a foundational study. We believe that there are neither ethical aspects nor future societal consequences that should be discussed at the current state of our work. Unsupervised criteria paired with a spatio-temporal filtering can potentially lead to the development of more robust features to describe visual information. In particular, the outcomes of this work could help in designing improved neural models, capable of extracting relevant information from a continuous video stream, from the same areas that attract the human gaze.

## Acknowledgments and Disclosure of Funding

This work was partly supported by the PRIN 2017 project RexLearn, funded by the Italian Ministry of Education, University and Research (grant no. 2017TWNMH2)

## Footnotes

[2]In this paper we changed the notation w.r.t. [3] in order to simplify the description of our approach.

[3]We removed the time index to simplify the notation. We will do it occasionally also in the rest of the paper.

[4]Since we are dealing with convolutional feature a realization of the random variable $X$ is specified by the coordinates of a point $x \in R$, the value of the temporal instant $t$ and the value of the video $u(t)$.

[5]When selecting a log in base $m$, the MI is in $[0, 1]$, that is what we will assume in the rest of the paper.

[6]Probabilistic normalization must be enforced after every update of $s(t)$.

[7]We decided to focus on a FOA model that is specifically designed to generate scanpaths, hence temporal sequences of fixation points, from a state-of-the art ODE-based formulation, and not a saliency map such as in [7] and other approaches in literature, making it well paired with our learning scheme. In our case, an independent gravitational process guides the visual gaze.

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
