[Supplementary Material]

# Supplementary Material

## A   Proof of Theorem 1

In order to prove Theorem 1 we first describe a technical hypothesis on the potential $U$. In detail, for all $\delta$ positive there exists two positive integrable functions $c_\delta(t)$ and $\kappa_\delta(t)$ such that for every $z \in \mathbb{R}^n$ and for all $t \in [0, T]$ we have

$$|\nabla U(z,t)| \leq \delta(U(z,t) + |z|^2) + c_\delta(t), \quad |\partial_t U(z,t)| \leq \delta(U(z,t) + |z|^2) + \kappa_\delta(t). \quad (1)$$

Notice that here, in order to simplify the notation, we use the same symbol for $U$ and for $\hat{U}(z,t) := U(z, u(t))$. We will also denote with $w_\varepsilon$ the solution of problem (5).

As it is also remarked below the proof articulates as follow: first of all we asses the convergence of $w_\varepsilon \to w$ by compactness arguments, basically by performing an estimate on the solution $w_\varepsilon$; then the uniform estimate on the $L^2$ norm of $\dot{w}_\varepsilon$ is used to check that the limit $w$ actually solves the problem (6).

*Proof.* The proof of this theorem follows the spirit of Theorem 4.2 of [26]. We will start with an uniform (in $\varepsilon$) estimate of $\|\dot{w}_\varepsilon\|_{L^2}^2$ and then we will use this estimate in weak form of the Euler equation to show the convergence of $w_\varepsilon$ to the solution of (6). We will prove the theorem in the case $\alpha > 0$ and $\beta = 0$.

*Uniform Estimate.* Start form the differential equation in (5) and scalar multiply it by $(w'_\varepsilon - w^1)$:

$$\varepsilon^2 \alpha w_\varepsilon^{(4)} \cdot (w'_\varepsilon - w^1) - 2\varepsilon\alpha w^{(3)} \cdot (w'_\varepsilon - w^1) + \alpha \ddot{w} \cdot (w'_\varepsilon - w^1) + \nabla U \cdot (w'_\varepsilon - w^1) = 0,$$

then integrate this equation on the interval $(0, t)$, and using the boundary conditions (5) integrate by parts to obtain

$$\varepsilon^2 \alpha w_\varepsilon^{(3)}(t) \cdot (w'_\varepsilon - w^1) - \frac{\varepsilon^2 \alpha}{2}|\ddot{w}_\varepsilon(t)|^2 + \frac{\varepsilon^2 \alpha}{2}|\ddot{w}_\varepsilon(0)|^2$$

$$-2\varepsilon\alpha w_\varepsilon^{(3)}(t) \cdot (\dot{w}_\varepsilon(t) - w^1) + 2\varepsilon\alpha \int_0^t |\dot{w}_\varepsilon(s)|^2 \, ds + \frac{\alpha}{2}|\dot{w}_\varepsilon(t) - w^1|^2$$

$$+U(w_\varepsilon(t), t) - U(w^0, 0) - \int_0^t \nabla U(w_\varepsilon(s), s) \cdot w^1 \, ds - \int_0^t \partial_t U(w_\varepsilon(s), s) \, ds.$$

Now let us integrate this equality again in the interval $(0, T)$, therefore obtaining

$$\left(2\varepsilon - \frac{3}{2}\varepsilon^2\right)\int_0^T \alpha|\ddot{w}_\varepsilon(s)| \, ds + \frac{\varepsilon^2(1+T)}{2}\alpha|\ddot{w}_\varepsilon(0)| + \left(\frac{1}{2} - \varepsilon\right)\alpha|\dot{w}_\varepsilon(T) - w^1|^2$$

$$+2\varepsilon\alpha \int_0^T \int_0^\tau \ddot{w}_\varepsilon(s) \, ds d\tau + \frac{\alpha}{2}\int_0^T |\dot{w}_\varepsilon(s) - w^1|^2 \, ds + U(w_\varepsilon(T), T)$$

$$+\int_0^T U(w_\varepsilon(s), s) \, ds = \int_0^T \nabla U(w_\varepsilon(s), s) \cdot w^1 + \int_0^T \int_0^\tau \nabla U(w_\varepsilon(s), s) \cdot w^1 \, ds d\tau$$

$$+(1+T)U(w^0, 0) + \int_0^T \int_0^\tau \partial_t U(w_\varepsilon(s), s) \, ds d\tau.$$

Now we can take all the positive (for $\varepsilon$ small enough) terms to the right hand side to obtain

$$\frac{\alpha}{2}\int_0^T |\dot{w}_\varepsilon - w^1|^2 \, dt + \int_0^T U(w_\varepsilon(t), t) \, dt \leq (1+T)U(w^0, 0)$$

$$+ (1+T)|w^1|\int_0^T |\nabla U(w_\varepsilon(t), t)| \, dt$$

$$+ T\int_0^T |\partial_t U(w_\varepsilon(t), t)| \, dt.$$

Now using Eq. (1) we can choose $\delta$ to further reduce this inequality down to

$$\frac{\alpha}{2} \int_0^T |\dot{w}_\varepsilon - w^1|^2 \, dt + \int_0^T U(w_\varepsilon(t), t) \, dt \leq c(T) + C(T) \int_0^T |w_\varepsilon(t)|^2 \, dt, \qquad (2)$$

where $c(T)$ and $C(T)$ are constant with respect to the parameter $\varepsilon$. Using Peter-Paul inequality we have that $|\dot{w}_\varepsilon - w^1|^2 \geq (1 - \eta')|\dot{w}_\varepsilon|^2 + (1 - 1/\eta')|w^1|^2$ for all $\eta' > 0$. Similarly since $w_\varepsilon \in H^2$, we can write $w_\varepsilon(t) = w^0 + \int_0^t \dot{w}_\varepsilon$ and using Peter-Paul and Cauchy-Schwartz we also end up with the estimate $|w_\varepsilon - w^0| \geq (1 - \eta)|w_\varepsilon| + (1 - 1/\eta)|w^0|$ for all $\eta > 0$, which implies

$$\int_0^T |w_\varepsilon(t)|^2 \, dt \leq T \frac{1/\eta - 1}{1 - \eta} |w^0|^2 + \frac{T^2}{1 - \eta} \int_0^T |\dot{w}_\varepsilon(t)|^2 \, dt. \qquad (3)$$

Putting together Eq. (2) and (3) we finally obtain the wanted uniform bound $\alpha \|\dot{w}_\varepsilon\|_{L^2} \leq k(T)$, where $k(T)$ is a constant with respect to the parameter $\varepsilon$.

*Convergence.* Once we have this uniform bound we can complete the proof by arguing along the very same lines of the proof of Section 3.2 of [26] to obtain the thesis. $\qquad\qquad\square$