[Reviews · NeurIPS 2020]

Review 1

Summary and Contributions: his paper investigates unsupervised learning from a continuous stream of visual input, with a visual selective attention mechanism that allows saccade-like dynamics. The method attempts to approximately maximize mutual information between the pixel-wise predictions of a deep network and the visual input stream. Experimental results show that focal attention reminiscent of human saccade choices improve mutual information between the visual stream and the network's output.

Strengths: This paper addresses an important problem: how can a system learn useful representations truly online, from a long sequence of video input. The experiments show that the proposed method achieves higher mutual information with the visual input stream compared to the more naive models considered. ------Update after rebuttal------ The theoretical framework could be useful to the community, but the empirical evaluation still seems anecdotal and hard to place in the broader context of work in the area.

Weaknesses: The evaluation shows that the proposed attention mechanism improves the mutual information metric defined in the paper, but this is not linked back to a clear functional benefit. This makes the significance of the MI improvements hard to interpret. Are the resulting features better able to perform some task of interest? Demonstrating this would strengthen the paper. The paper introduces fourth order dynamics, and spends considerable time simplifying this to 2nd order dynamics. It is unclear what this adds to the presentation, and it may be more straightforward to simply state what the implemented model is directly. The experiments chose hyperparameters to maximize the mutual information extracted by each algorithm, and it is not clear from the text whether this optimization was performed on a validation set or not. It seems possible therefore that some of the results could be due to overfitting the hyperparameters to the test dataset. The paper could be improved with a more complete discussion of hyperparameter selection. The evaluation videos could be better motivated. They are quite specific (exactly three videos), and the paper would be improved by discussing the significance of looping the videos. How many loops were necessary to generate 105k frames for the Carpark/Call streams, for instance? It is hard to tell from these specific examples how robust the results are.

Correctness: The theoretical framework as initially introduced requires many approximations and simplifications in practice, but these are relatively well-motivated. The experiments appear to be correct, and admirably, full code is provided.

Clarity: I found the paper somewhat difficult to follow. I think the presentation of the 4th order dynamics could be considerably shortened, with the exposition focusing on exactly what algorithm was implemented in the experiments.

Relation to Prior Work: Prior work is clearly discussed, and this paper's contributions are clear.

Reproducibility: Yes

Additional Feedback:


Review 2

Summary and Contributions: Presents a detailed mathematical description of a method for optimizing information transfer of an attentive visual system, based on the least action principle.

Strengths: (see the Weaknesses section for further context) I think the main contribution of the paper is the very detailed and novel mathematical specification and solution of the variational learning problem (resulting in what the paper calls the "Cognitive Action Laws").

Weaknesses: The approach is well-founded and the derivations are solid, but the main problem with the paper is that the general approach and motivating ideas are very similar to the free energy principle ideas of Karl Friston and his co-workers. The authors need to contrast their work with that of Friston and point out the novel contributions that are being made. in the experiment section, comparison should be made to the input/output mutual information at fixation/attention locations generated by other state-of-the-art video attention algorithms e.g. such as the one in: Cornia, M., Baraldi, L., Serra, G., & Cucchiara, R. (2018). Predicting human eye fixations via an lstm-based saliency attentive model. IEEE Transactions on Image Processing, 27(10), 5142-5154.

Correctness: Yes

Clarity: Yes, the paper is clear, although there are many grammatical errors. These errors do not affect the readability of the paper, however.

Relation to Prior Work: I am surprised to see that a paper proposing a method based on the principle of least action does not cite the work of Karl Friston, who is the most well-known proponent of this idea as it relates to perception and cognition. Friston, Karl. "The free-energy principle: a unified brain theory?." Nature reviews neuroscience 11.2 (2010): 127-138. Karl, Friston. “A Free Energy Principle for Biological Systems.” Entropy (Basel, Switzerland) vol. 14,11 (2012): 2100-2121. doi:10.3390/e14112100 (from the abstract of the latter paper: "We motivate a solution using a principle of least action based on variational free energy (from statistical physics) and establish the conditions under which it is formally equivalent to the information bottleneck method.") Friston and co-workers have applied his ideas to modeling attention. For example, in the following paper they state: "We have suggested recently that perception is the inference about causes of sensory inputs and attention is the inference about the uncertainty (precision) of those causes (Friston, 2009). This places attention in the larger context of perceptual inference under uncertainty" Feldman, H., & Friston, K. (2010). Attention, uncertainty, and free-energy. Frontiers in human neuroscience, 4, 215. See also: Schwartenbeck, P., FitzGerald, T., Dolan, R., & Friston, K. (2013). Exploration, novelty, surprise, and free energy minimization. Frontiers in psychology, 4, 710. The paper should compare the proposed focus of attention modeling approach with prior methods employing mutual information, such as that of Bruce and Tsotsos: Bruce, N., & Tsotsos, J. (2006). Saliency based on information maximization. In Advances in neural information processing systems (pp. 155-162).

Reproducibility: Yes

Additional Feedback: I like the paper mainly because it provides a different approach to the problem. Although the underlying motivating idea is similar to others, such as the Friston free energy paradigm, they have a different way of attacking the problem. So I think it is useful to others building such models. To me, the rebuttal didn't really seem to answer any of the reviewers' main questions, so I don't know if any minds will be changed here. I still think the paper is worth accepting.


Review 3

Summary and Contributions: The paper propose an interesting combination between Lagrangian (classical mechanics) formulation of temporal (streaming) learning over time with Information theoretic view of human like attention mechanisms. The Lagrangian approach leads to a second order ODE models on top of which the authors propose a Mutual Information maximization to capature attention. The learning over time formulation is analogous to the Lagrangian formulation of classical mechanics, with a kinetic and potential energy terms which are functions of the generalized vector time flow (generalized canonical coordinates) wand their forests 2 derivatives. They added a disipative exponentially decaying term which can include loss of energy in the dynamics, to allow for non conservatives flows. The learning algorithm try to estimate the optimal potential function - which can be time dependent, and a few discrete parameters of the kinetic term. On top of that formalism, they add an Information or entropy measure which is a functional of the probability density function of these trajectories (something that in principle requires to solve a Focker-Planck like equation). Assuming that this can be done, they use this spatio-temporal density to calculate the mutual information between different times along the dynamics of this flow. This rather involved formalism is applied to several videos by fitting a time dependent potential to the dynamics of the pixels in each video and then estimate the attention (salient) points in each of the scene. They also relate or compare it to the neuroscience of the retina.

Strengths: The paper is inspiring by its physics-like approach to several hard problems, trajectory learning in a Lagrangian framework, flow density estimation, mutual information changes along the flow and its relationship with attention mechanisms. If convincingly controlling the algorithms involved - it can be a very interesting contribution. Not so much to the NIPS audience though, as the emphasis is on the physics like formulation.

Weaknesses: This is a very ambitious paper that tries to combine several novel and computationally difficult tasks in one framework and apply them to real data with claims to be relevant to neuroscience. The main problem is that these difficult tasks,: learning potentials for the classical mechanics of trajectories, ensemble density estimation of these trajectories, mutual information estimation for this ensemble and then extraction the attention points from these estimates, are too much for one short paper. The paper didn’t convince me that any of this difficult problems is satisfactory solved algorithmically here, and their combination on the real data, which gives interesting comparative values in these datasets, but look anecdotal and inconclusive to me. The paper completely ignores other algorithms for learning temporal flows, like RL or DeepRL, and does not specify well enough the algorithmic issues. It is not clear how to compare this approach to any standard method and what is the real benefit of the proposed measure of attention estimation, to cognitive processing in general.

Correctness: The main theorem (Thm 1) and its proof in the supplementary seem correct, but no learning algorithms are given in the paper. I couldn’t understand how exactly the trajectory density estimation, on which the mutual information estimation is based, is actually calculated. Solving FP like equation in high dimensions is nuitoriously difficult without some parametric assumptions. What is actiually assumed about the unknown potential U? How is it learned from the data? These are crucial details that are missing from this short paper, which make the result difficult to evaluate or reproduce.

Clarity: The paper is clearly written and can be understood by people familiar with the concepts of Lagrangian dynamics, entropy and mutual information, agttantion mechansism , and this physics style math. Otherwise, the paper is inaccessible.

Relation to Prior Work: The references are limited to this very specific line of research with its several threads. I missed a comparisons to other dynamic learning methods and algorithms, such as Deep RL, InfoRL, and their extensions. While the submission contains the video data and Peyton code and I suppose the results can be reproduced by running the code on this data, it isn’t clear to me what happens on other data and what is the meaning and how to interpret the main numerical results in Table 2.

Reproducibility: Yes

Additional Feedback:


Review 4

Summary and Contributions: This paper deals with a problem of unsupervised statistical modelling from time-series signals and proposes a method based on mutual information maximisation. The proposed method is originated from previous papers presented by a specific author group and is extended to spatio-temporal signals. The validity of the proposed method is evaluated with simple video signals.

Strengths: 1. This paper has a strong mathematical background that have been developed by the same author group.

Weaknesses: 1. Low clarity. Honestly speaking, I could not follow the discussion in the current manuscript at all, partly due to the lack of my knowledge for the previous methods presented by the same author group. 2. If my understanding is correct, the current manuscript does not contain any experimental comparisons with other previous methods related to statistical modelling of video signals.

Correctness: Mathematical discussions seem to be no problem, as far as I understand. However, there exists a big logical gap between the motivation presented in the abstract and the methodology.

Clarity: Clearly no. The current manuscript is not self-contained and requires extensive knowledge for the methods proposed by a specific author group. We can find a lot of "See ... for the detail" and motivations and meanings of mathematical expressions have been totally skipped in this paper.

Relation to Prior Work: I think that it is no problem, in terms of follow-ups of a specific author group.

Reproducibility: No

Additional Feedback:

[Author Response · NeurIPS 2020]

**Paper Title:** Focus of Attention Improves Information Transfer in Visual Features (Paper ID 9698)

We thank the Reviewers for their comments and their positive feedback both on the problem we decided to tackle and on the approach we followed! We did our best to answer the provided questions and to reply to some comments.

**Reviewer #1.** "*Are the resulting features better able to perform some task of interest?*" We considered the problem of maximizing the Mutual Information (MI) index in order to avoid focusing on a specific task, thus having an unsupervised learning criterion. Of course, the resulting features are capturing precious information about the context around each pixel, so they might help in some downstream tasks. We tried to preliminary face a semantic labeling task, and the MI-based features were helping in case of small/sparse supervision.
"*The paper introduces fourth order dynamics, and spends considerable time simplifying this to 2nd order dynamics. It is unclear what this adds to the presentation [...]*" We aim at providing the precise theoretical groundings from which what we implemented comes from. The 2nd order dynamics of Eq. (6) draws much of its motivation from Theorem 1, which is based on a particular limit of the 4th order ODEs of Eq. (5) – stationary points of functional (4). Notice that, as far as we know, there is indeed no direct way to derive Eq. (6) from a simpler functional than (4) while retaining only the correct initial conditions (because of the inherent difficulty of deriving hyperbolic problems from variational principles). In turn, functional (4) has strong analogies with functional (2), already studied in related work, that is the one from which we start the presentation, creating a clean connection to the existing literature.
"*The experiments chose hyperparameters to maximize the mutual information extracted by each algorithm, and it is not clear from the text whether this optimization was performed on a validation set or not*"
We selected the parameters that were leading to the largest MI index at the end of the learning stage, Eq. (8), and not on the test data (thanks for this comment – we were not explicitly mentioning what data were used).
"*The evaluation videos could be better motivated. They are quite specific (exactly three videos), and the paper would be improved by discussing the significance of looping the videos. How many loops were necessary [...]*" We selected videos that naturally represent repetitive contents, so that they could be looped without introducing evident scene changes (a video call from a laptop, a camera recording cars in a parking lot, static digits). Contents and events are heterogeneous among the videos, and we believe that they are able to resemble the continual life-long flow of information hitting the human eyes. The CARPARK and CALL videos are composed of 1259 and 2386 frames, respectively – we looped them $\approx 83$ and $\approx 44$ times, respectively.

**Reviewer #2.** "*The authors need to contrast their work with that of Friston and point out the novel contributions that are being made [...]. In the experiment [...] fixation/attention locations generated by other state-of-the-art video attention algorithms*" We thank the Reviewer for providing as references the work by Frinston, that we will certainly discuss in case of acceptance! Shortly, while definitely sharing several connections, we believe that is mostly the context of this work (mutual information in video streams/visual features with deep nets paired with human-like focus of attention) that represent a novel contribution, aimed at improving time-oriented machine learning systems exploiting a model of human attention. Regarding the attention models suggested by the Reviewer, we will consider them as well, even if here we decided to focus on a FOA model that is specifically designed to generate scanpaths, hence temporal sequences of fixation points, from a SOTA ODE-based formulation and not a single saliency map (such as in Cornia et al. and other approaches in literature), making it well paired with our learning scheme. We experimented different FOA model parameters, generating different (valid) scanpaths, getting similar results. In the seminal bottom-up model by Bruce and Tsotsos attention emerges maximizing the Self-Information of each local image patch. Conversely, in our case an independent gravitational process guides the visual gaze, which we show to favour the information transfer.

**Reviewer #4.** "*(Thm 1) and its proof in the supplementary seem correct, but no learning algorithms are given [...]*" Instead of the classical gradient-descent-based update rule $\dot{w} = -\nabla U(w(t), t)$ our model exploits the second order equation of Eq. (6). We are integrating the equation with the Euler method (we have to specify this in the experimental section, thanks for this comment!).
"*Couldn't understand how exactly the trajectory density estimation, on which the mutual information estimation is based, is actually calculated. Solving FP like equation in high dimensions is nuitoriously difficult without some parametric assumptions.*" We compared different choices for the function $g$ which defines the probability measure $\mu$. As we briefly discussed towards the end of Section 3 we considered the case of uniform density over the frame, and the case in which g is a peaked function on the trajectory $t \mapsto a(t)$ which in turn is estimated using the approach proposed in [29].
"*What is actually assumed about the unknown potential U? How is it learned from the data?*" In this paper, the potential $U$ is equivalent to the negative mutual information (see the details in Section (2)), that depends on the output of a Deep Network composed of convolutional layers – whose filters are learned from data.

**Reviewer #5.** "*I could not follow the discussion in the current manuscript at all, partly due to the lack of my knowledge for the previous methods [...]*" The paper is about information transfer in vision problems, merging in a clean formulation SOTA focus of attention models and learning over time. We do agree that it covers several topics and we provided precise references to get more details about all of them. Notice that we are also sharing the code to help reproducibility.

[Meta-Review · NeurIPS 2020]

This paper received mixed reviews: R2 & R4 recommended accept (score 7), R1 recommended weak reject (score 5), and R5 recommended a clear reject (score 3). However, R5 mentioned that the reviewer is unfamiliar with the topic as well as the prior work in this domain. Therefore, I am discarding R5's review. R2 & R4 acknowledged that this paper makes a strong theoretical contribution to optimizing information transfer of human-like visual attention mechanisms; they praised the mathematical construction is "well-founded," "solid," and "inspiring" (R1 also agreed with this after the rebuttal/discussion phase). Perhaps one weakness of this paper is insufficient empirical evidence. All three reviewers (R1, R2 & R4) raised similar concerns on this point. However, given strong theoretical justifications/analyses provided in the paper (e.g., the arduous derivation of the fourth-order dynamics to justify the second-order approximation, the ambitious attempt at combining several computationally difficult tasks in one framework, etc.) I think this is of sufficient quality to be presented at NeurIPS.